# Semimonthly oscillation observed in the start time of equatorial plasma bubbles

Igo Paulino[1], Ana Roberta Paulino[2], Ricardo Y. C. Cueva[3], Ebenezer Agyei-Yeboah[4], Ricardo Arlen Buriti[1], Hisao Takahashi[5], Cristiano Max Wrasse[5], Angela M. Santos[5], Amauri Fragoso de Medeiros[1], and Inez S. Batista[5]

[1]Unidade Acadêmica de Física, Universidade Federal de Campina Grande, Campina Grande, Brazil
[2]Departamento de Física, Universidade Estadual da Paraíba, Campina Grande, Brazil
[3]Departamento de Física, Universidade Estadual do Maranhão, São Luís, Brazil
[4]Instituto de Pesquisa e Desenvolvimento, Universidade do Vale do Paraíba, São José dos Campos, Brazil
[5]Divisão de Aeronomia, Instituto Nacional de Pesquisas Espaciais, São José dos Campos, Brazil

**Correspondence:** Igo Paulino (igo.paulino@df.ufcg.edu.br)

**Abstract.**

Using airglow data from an all sky imager deployed at São João do Cariri (7.4$^o$S, 36.5$^o$W), the start time of equatorial plasma bubbles was studied in order to investigate the day-to-day variability of this phenomenon. Data from a period over 10 years was analysed from 2000 to 2010. Semimonthly oscillations were clearly observed in the start time of plasma bubbles from Oi6300 airglow images during this period of observation and four case studies (September 2003, September-October 2005, November 2005 and January 2008) were chosen to show in details this kind of modulation. Since the airglow measurements are not continuous in time, more than one cycle of oscillation in the start time of plasma bubbles cannot be observed from these data. Thus, data from a digisonde at São Luís (2.6$^o$S, 44.2$^o$W) in November 2005 were used to corroborate the results. Technical/climate issues did not allow to observe the semimonthly oscillations simultaneously by the two instruments, but from October to November 2005 there was a predominance of this oscillation in the start time of the irregularities over Brazil. Besides, statistical analysis for the data in the whole period of observation has shown that the lunar tide, which has semimonthly variability, is likely the main forcing for the semimonthly oscillation in the start time of equatorial plasma bubbles. The presence of this oscillation can contribute to the day-to-day variability of equatorial plasma bubbles.

**Keywords:** Spread-F, Plasma bubble, Semimonthly oscillation, Lunar tide, 16d planetary wave.

## 1 Introduction

Equatorial plasma bubbles (EPBs) are generated in the bottom side of the F-region in the equatorial ionosphere when there is an unstable F-layer. They generally occur after the pre-reversal enhancement (PRE), after sunset. The pre-reversal enhancement consists of an increase in the eastward electric field, before its reversal to the west, which causes an enhancement in the F layer vertical drift before the motion of the plasma be downward reverted. The main mechanism used to explain the development of the EPBs is the Rayleigh-Taylor (RT) instability. According to this theory, the RT growth rate is inversely proportional to the

collision frequency between the neutral and ionic particles and it is proportional to the plasma density gradient. Thus, when the PRE is strong, it becomes more probable for EPBs occur.

In addition, the RT instability process needs a seeding mechanism, which has been largely studied in the last decades. Some researchers have pointed out gravity waves as seeding to the EPB (e.g., Fritts et al., 2008; Abdu et al., 2009; Takahashi et al., 2009; Taori et al., 2011; Paulino et al., 2011). Other studies have marked the dynamics of post sunset vortex and PRE dynamics as enough to the EPB origin (e.g., Kudeki and Bhattacharyya, 1999; Kudeki et al., 2007; Eccles et al., 2015; Tsunoda et al., 2018; Huang, 2018). The thermospheric neutral wind system and the associated electrodynamics have also been proposed as sufficient to the EPB appearance as well (e.g., Saito and Maruyama, 2009). Influences of magnetic storms and large scale waves have also been reported as important mechanism to the day-to-day variability of EPBs (e.g., Abalde et al., 2009; Huang et al., 2013).

Actually, observations have shown that there is a strong day-to-day variability of the EPB occurrence and development (e.g., Carter et al., 2014; Abdu, 2019) and it is a topic of current research. There is evidence of planetary waves acting in the neutral winds and consequently changing the background condition of the atmosphere (e.g., Forbes, 1996; Takahashi et al., 2006; Abdu and Brum, 2009; Chang et al., 2010; Onohara et al., 2013; Zhu et al., 2017).

Stening and Fejer (2001) published the first work proposing the influence of lunar tides in the probability of occurrence of EPBs. It is well known that the main component of the lunar tides has a semimonthly oscillation. Based on these factors, the present work shows, for the first time, that there are semimonthly oscillations statistically significant in the start time of EPBs observed by airglow images throughout the period of observations. Besides, these oscillations follow the Moon phases. These results can indicate strong evidences of the lunar semidiurnal tide modulating the wind system in the F region and consequently it is driving the time of generation of Spread-F.

## 2 Data Analysis

Airglow measurements of the OI 630.0 nm (OI6300) have been recorded at São João do Cariri ($7.4^{o}$S, $36.5^{o}$W) since September 2000. In this investigation, data from September 2000 to December 2010 were used, which corresponds to the first generation of the all sky imager deployed in this observatory.

The all sky imager is composed by a fish eye lens, a telecentric set of lens, a filter wheel, a set of lens to reconstruct the image, a Charge Coupled Device (CCD) chip and a cooling system. This instrument has a field of view of $180^{o}$ of the sky. Further details of this imager have been published elsewhere (e.g., Paulino et al., 2016). Airglow images of the OI6300 were taken by about 15 days around the New Moon with integration time of 90 s. Depending on the mode of operation, images can have 2-4 min of temporal resolution. The start and end times can be extracted directly from the image header after observing the appearance or disappearance of the structures. The start time was defined as the time when the plasma bubbles appeared in the images. It generally occurs in the Northwest part of the images. After that, the plasma bubbles start their development and dynamics.

Figure 1 shows an example of the determination of the start time of EPBs on 27 January 2001. The supplementary short movie can help one to identify the time, in which the plasma bubbles start to extend to the southern part of the images.

Corroborative data from a Digisonde Portable Sounder (DPS) installed at São Luís (2.6$^o$S, 44.2$^o$W) were also used to identify the time of maximum vertical drift of the F layer, which corresponds to the time of the pre-reversal enhancement. The DPS is an HF radar which operates in a sweeping mode in the frequency range from 1 to 30 MHz, generating one ionogram (graphic of frequency versus virtual height) each 10 min. Besides the ionogram the DPS also detects the doppler shifts of the irregularities, which can be used to calculate the drifts. Data collected in October 2003, October 2005 and November 2005 were investigated. However, semimonthly oscillation in the time of maximum vertical drift of the F layer were observed only in November 2005 and it will be discussed latter. Further details about the digisonde deployed at São Luís and the methodology of determination of the vertical plasma drifts from ionograms have been published elsewhere (e.g., Resende et al., 2019).

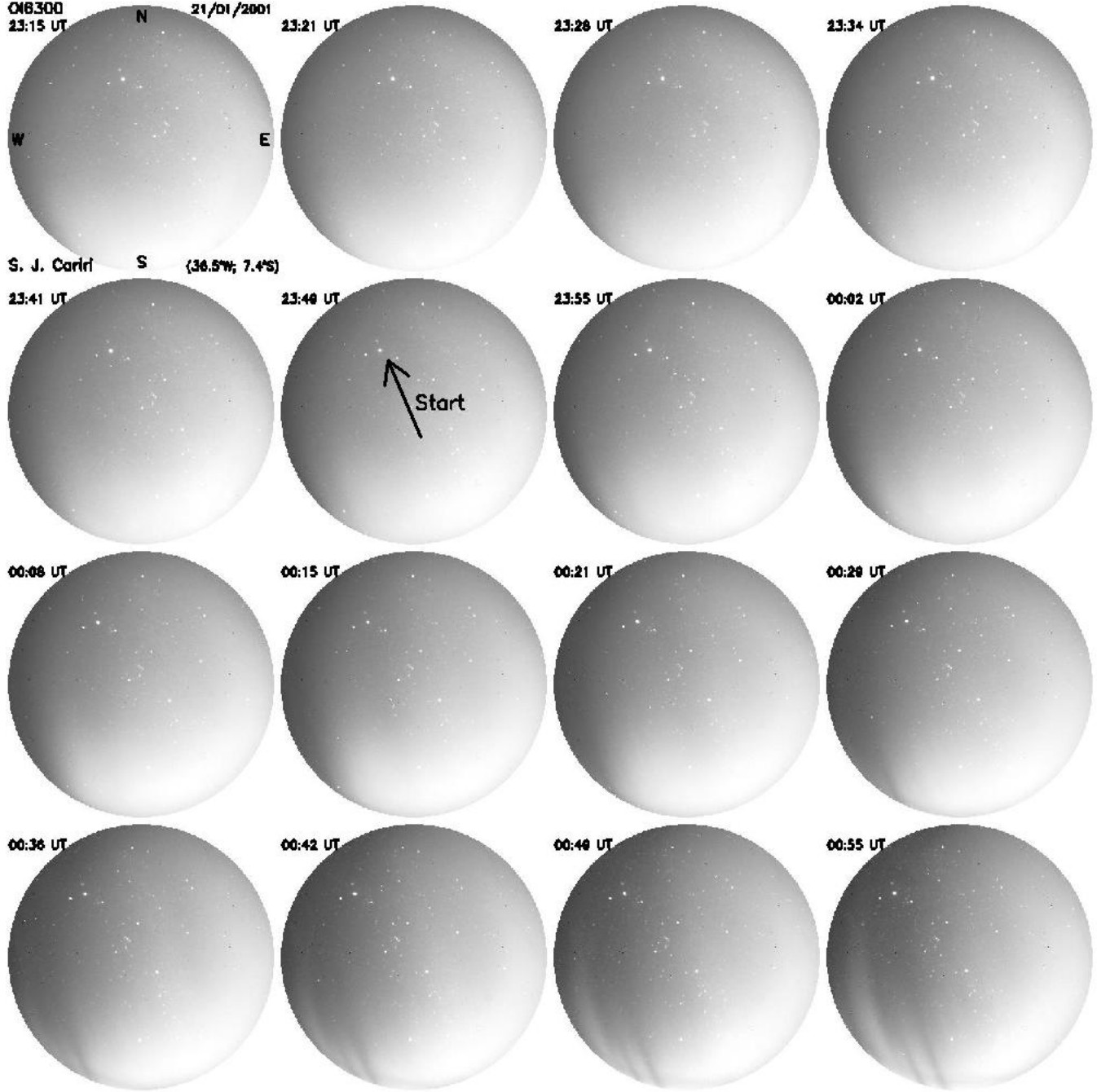

**Figure 1.** Sequence of OI6300 airglow images observed in São João do Cariri on 27 January 2001. One can observe the start time of the EPB on this night.

## 3 Results and Discussion

Figure 2(a) shows the evolution of the start time of the EPBs observed in September 2003 over São João do Cariri. The solid line represents the best fit for a periodicity of 14.5 days, the stars correspond to the exact time in which the plasma bubble appeared in the OI6300 images and the filled circle shows the New Moon time. In this case, one can see a good agreement of the fit line with the observation during a half cycle of the oscillation. The amplitude of this oscillation was calculated from the fitting as ∼52 min, i.e., there was a difference of ∼52 min in the start time of EPBs along the observed nights.

Figure 2(b) shows the best fit 14.5 days oscillation in the start time of EPBs observed from the later September to early October 2005. For the whole period of airglow observation, it was the best case study observed because it covers a full cycle of the oscillation. There was an amplitude of ∼37 min and the position of the New Moon was observed on 03 October 2005. The predominance of this oscillation in the start time of EPBs persists up to November 2005 as shown in Figure 2(c) with higher amplitude ∼70 min.

Similar results to September 2003 and November 2005 were found in January 2008 as one can see in Figure 2(d), inclusive the position of the New Moon in the cycle. The estimated amplitude was ∼45 min.

The results from Figure 2 indicate that the start time of EPBs was modulated by a semimonthly oscillation. Besides the results from Figure 2, there were other events that showed a tendency of the start time of EPBs follow the semimonthly periodicities. However, only few days, less than a half month, were observed and those results are not shown here. Additionally, long term statistical analysis will be discussed latter.

The present work concentrate the discussion on the cases in which, a half cycle could be observed. Semimonthly oscillations well known in the atmosphere are: (1) Quasi 16 days planetary waves and (2) Lunar semidiurnal tide.

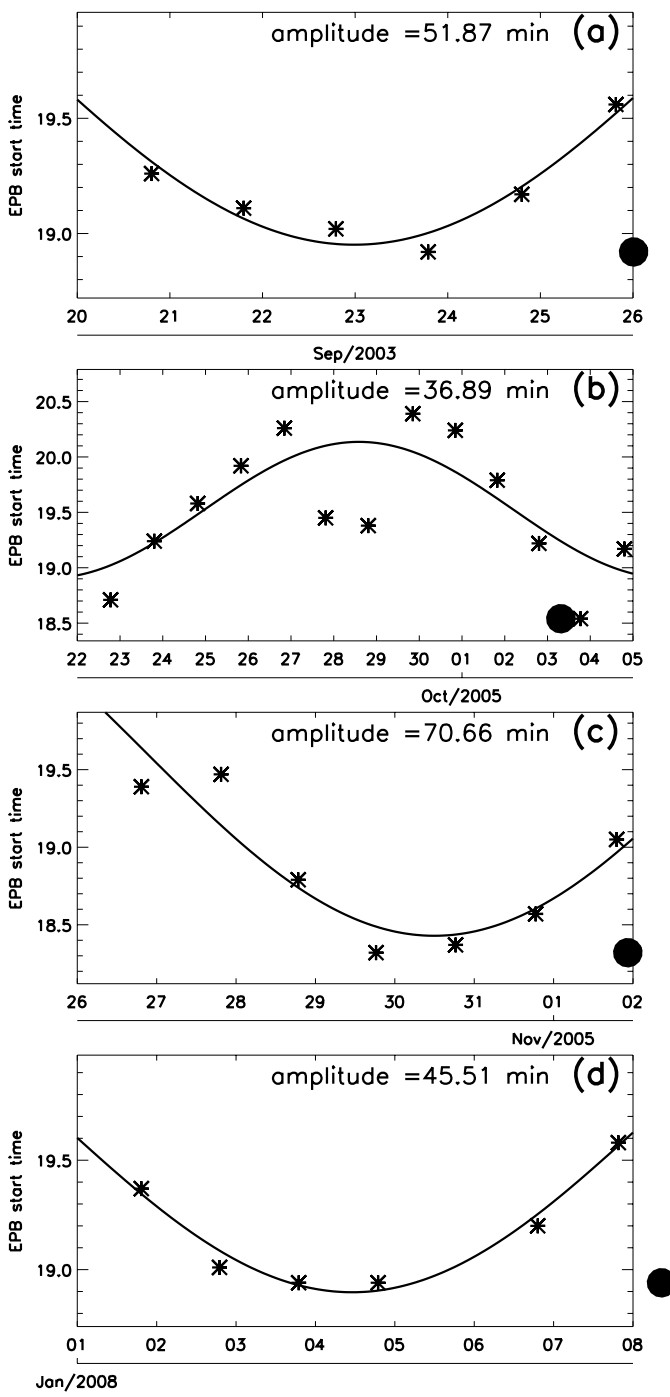

**Figure 2.** Start time of plasma bubble (stars) as function of time. Solid line represents the best fit to a sinusoidal oscillation with period of 14.5 days. The respective amplitudes are shown on the middle top of the panels. Panel (a) shows the results for September 2003. Panel (b) shows the results for September - October 2005. Panel (c) shows the results for October-November 2005. Panel (d) shows the results for January 2008. Filled circles indicate the New Moons.

Simulations have shown that the 16d planetary waves (PWs) have large amplitudes in the winter hemisphere at the lower levels of the atmosphere and high latitudes, but above the mesosphere, there is a penetration of this wave to the summer hemisphere, which allows that they can be observed in both hemispheres including in the equatorial region (Miyoshi, 1999).

Forbes and Leveroni (1992) have pointed out that 16d oscillation in the E and F-region could be connected by the upward propagation of Rossby wave from the winter stratosphere. Although, the 16d PW has a well defined seasonality in the lower atmosphere, according to the simulations, in the upper atmosphere the presence of this oscillation has been predicted to be more spread along the year (Miyoshi, 1999). It is also important to mention that the 16d oscillations were observed in the mesosphere and lower thermosphere from 85 to 100 km altitude in the equatorial region in the zonal wind during the period around the September equinox and solstices of 1994 (Luo et al., 2002), which coincides to the periods of observation of the results of Figure 2.

Lunar semidiurnal tides have been pointed out as important factor to the appearance and the start time of EPBs (e.g., Stening and Fejer, 2001). The main reason for the influence of the lunar tides in the EPB variability is the capability of the lunar tides propagate upward to high levels of the atmosphere and consequently to affect the pre-reversal enhancement (PRE) amplitude and time (Stening and Fejer, 2001). Another factor to be considered is the moon phase (New Moon) that coincides to zero position of the oscillation for all observed cases, including the case study observed from the DPS that will be shown ahead. The real mechanism that allows the lunar tides to act in the PRE is not well defined, but some works have pointed out as either the direct propagation to the bottom side of ionospheric F region (e.g., Evans, 1978; Forbes, 1982) or coupling of the E region dynamo to the F region (e.g., Immel et al., 2009; Eccles et al., 2011).

In order to corroborate the present results, data from the DPS deployed in São Luís have been used to investigate the occurrence of maximum vertical drifts. The main goal of these analysis is trying to observe more this kind of oscillation in other ionospheric parameter. Although the DPS operates continuously every day, i.e., the digisonde does not depend on the tropospheric weather conditions, only a half cycle of the oscillation could be observed in the used data.

Figure 3 shows the temporal evolution of the time of the maximum vertical drift observed from the DPS data in early November 2005. An amplitude of $\sim 46min$ was calculated, indicating that the PRE is sensible to the semimonthly oscillation as well. An important factor to these observations was that this oscillation acted in the ionosphere for a long period, since the start time of EPBs from later September (Figure 2b). There were simultaneous measurements of the start time of EPBs and the time of maximum vertical drifts in October 2005. However, the latter have not present reliable results for the semimonthly oscillation.

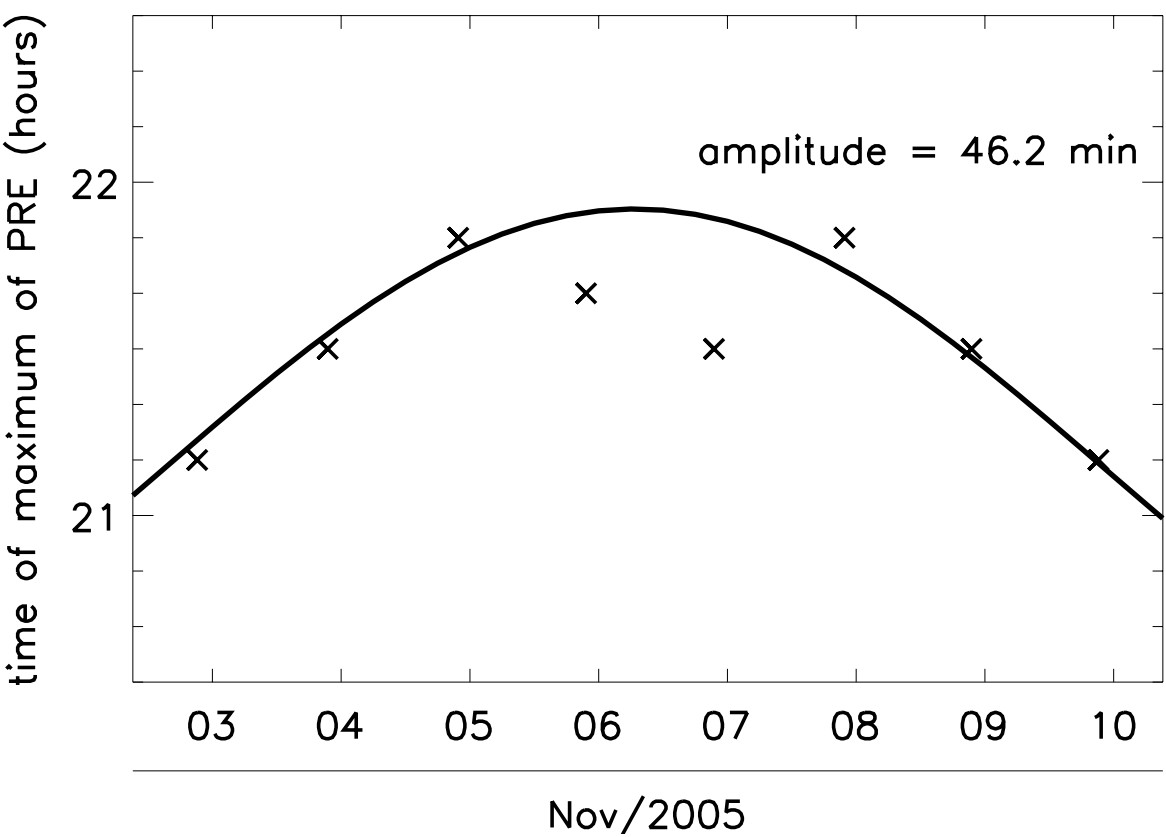

**Figure 3.** Same as Figure 2, but for time of maximum of vertical drift of the F layer.

Figures 2 and 3 show that the ionospheric parameter can be controlled by semimonthly oscillations. However, the strong day-to-day variability of the spread-F does not allow to observe this signature always. Another difficulty in the DPS data analysis was the algorithm does not give an exact start time of the oscillation, i.e., there was a temporal resolution of 10 min in this determination.

5    Although as the performed fit to the start time of EPBs and the fitting to the time of maximum vertical drifts of the F layer presented high amplitudes and very good agreements with the observation, only one case studied presented an almost full cycle (Figure 2b). Then, a statistical analysis was done in order to observe the relevance of this approach and how much frequent is the modulation of the semimonthly oscillation in the start time of EPBs. This analysis was performed considering the potential effect of the lunar tides in the ionosphere as simulated and discussed by (Stening and Fejer, 2001). In order to do that, a

10    methodology described by (Matsushita, 1967) has also been used.

Figure 4 shows hourly means of the the local start time of EPBs as a function of the lunar local time between 18:00 and 01:00, which corresponds to almost one cycle of the lunar tidal oscillation (12 hours) with maximum after 01:00, minimum

around 21:00 local lunar time and amplitude of ∼13.5 min. One can note that the scattering (error bars) change along the time. Assuming the standard deviation as a measurement of uncertainty, the correspondent $1\sigma$ level of uncertainties were ∼4.8 min in amplitude and ∼0.63 h in phase. According to the normal distribution, 68% of the points are between the $\pm 1\sigma$ level around the mean value. Furthermore, one can see a good agreement between the 12 hours oscillation and the data sets. This methodology has been also published in the work by Forbes et al. (2013). It is important to highlight that besides the well known temporal variability of the lunar tide, the present data carries influences of other geophysical variability in the start time of EPBs.

The lunar time was calculated as $\tau = t - \nu$, where, $t$ is the local solar time and $\nu$ is the age of the Moon, which depends on the phases of the Moon. Further details about the calculation of the lunar time can be found in Paulino et al. (2017) and references therein.

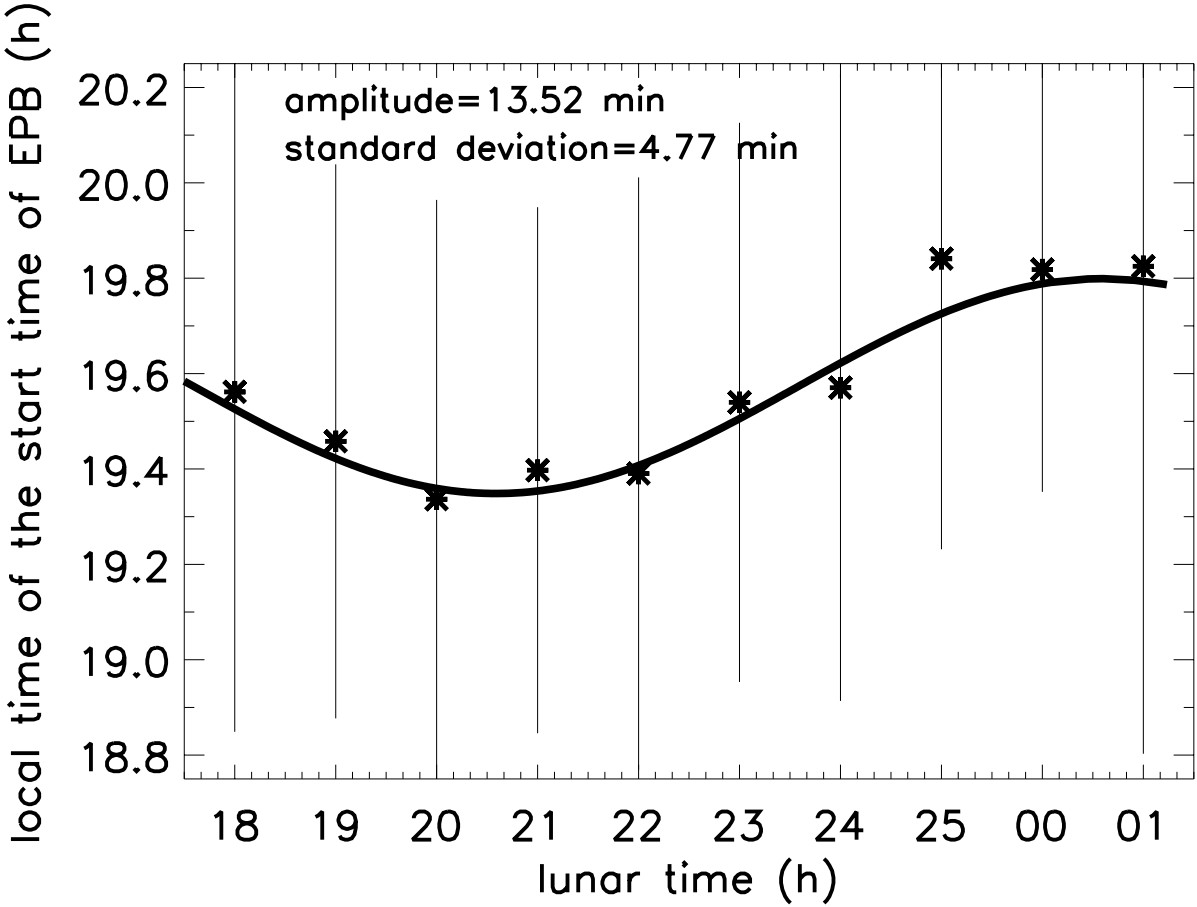

**Figure 4.** Lunar Semidiurnal $M_2$ fit (solid line) to the start time of EPBs observed from the airglow images. The error bars are the standard deviation of the data within each hourly bin.

From the results of Figure 4, it is clear that the semimonthly oscillation is always present in the start time of EPBs with a significante amplitude. It suggests that the lunar semidiurnal tide, which has a well defined semimonthly variation, has an important role in the time of occurrence of EPBs. Previous studies have pointed out that the lunar semidiurnal tide can modulated ionospheric parameters such as the height and critical frequency of the F layer, PRE drifts, etc. The present results strongly suggest that the generation of EPBs are affected as well. Further analysis of the start time of Spread-F, using radar measurements will be important in the advances of the knowledge of the day-to-day variability of EPBs.

## 4  Summary

Using almost one solar cycle of data from OI630 airglow images, semimonthly oscillations in the start time of EPBs were observed and the results are summarized as follow:

- Four periods of airglow observation showed amplitudes higher than 36 min in the start time of EPBs for 14.5 days oscillation, three periods of observations (September 2003, October 2005 and January 2008) revealed good fit for half cycle and the another case (September 2005) showed and complete cycle;

- DPS measurements from São Luís showed semimonthly oscillation in the maximum vertical drifts of the F region related to the PRE;

- Statistical analysis in the whole period of observations of EPBs in the airglow images revealed that semimonthly oscillations are always present in the start time of EPBs, when the lunar time was considered. Thus, it strongly suggest that the lunar semidiurnal tide has an important role the start time of EPBs.

The present results indicate that one semimonthly dynamical structure can control either the start time or the amplitude of the PRE that can consequently produce EPBs. These results must contribute to understanding the day-to-day variability of equatorial plasma bubbles. However, the results show that besides the semimonthly oscillations, other phenomena are important to the day-to-day variability occurrence of EPBs since this oscillations is not dominant in the whole period of observation. Regarding to the agents that are causing this oscillation, further investigation are necessary, however, semidiurnal lunar tides appeared as an important phenomenon to the time of the appearance of EPBs.

*Data availability.* All sky image data can be requested from either the Aerolume (UFCG) or Lume (INPE) Groups to the e-mail address to the first author of the manuscript. DPS ionograms can be requested to Dr. Inez S. Batista (inez,batista@inpe.br)

*Author contributions.* IP has written the manuscript and made most of the airglow analysis. ARP has discussed the semimonthly oscillation due to lunar tides and 16d planetary waves. RYCC has contributed to the discussion on the start time of EPBs. EA-Y has reduced the whole image data calculating the start time of EPBs. RAB has contributed to run the experiments in São João do Cariri and help with the analysis.

HT has contributed to the discussion of 16d oscillation. AMS has evaluated the time of maximum vertical drifts of the F layer. AFM has provide some computing codes to work with the OI6300 airglow images. ISB has provided the DPS data for analysis.

*Competing interests.* The authors declare that they do not have competing interests;

*Acknowledgements.* I. Paulino and C. M Wrasse thank to Conselho Nacional de Desenvolvimento Científico e Tecnológico (CNPq) for the financial support (303511/2017-6, 307653/2017-0). A. R. Paulino thanks to the Coordenação de Aperfeiçoamento de Pessoal de Nínel Superior (CAPES) for the scholarship and CNPq by the grant (# 460624/2014-8). Ângela M. Santos acknowledges the Fundação de Amparo à Pesquisa do Estado de São Paulo - FAPESP for the financial support under grant 2015/25357-4.

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
