# Peer review of "Semimonthly oscillation observed in the start time of equatorial plasma bubbles"

_Annales Geophysicae, 2019_

## Referee Comment (RC1) · Anonymous Referee #1 · 4 Jun 2019

In this study, the authors investigated the start time of equatorial spread-F (ESF) by using the long term dataset (2000-2010) from an all-sky airglow imager and a coherent backscatter radar at the two stations Sao Joao do Cariri and Sao Luis, respectively. They reported that the semimonthly oscillations were seen in the start time of ESF during the periods September 2003, September-November 2005, January and November 2008. It was suggested that the ESF semimonthly oscillations could be associated with the 16-day planetary waves and/or lunar semidiurnal tides which affected the pre-reversal enhancement of eastward electric field (PRE) around sunset. The result of ESF semimonthly oscillation is interesting and deserves to be published.

Some comments, 1. Although the authors stated that a long term dataset was used in this study, it is not clear how often the semimonthly oscillation was observed. Whereas

the weather could cause lack of data from all-sky airglow imager, VHF radar should not be affected. It would be better if one more figure can be included to provide both the periods with data (all-sky airglow imager and VHF coherent, respectively), and the periods when semimonthly oscillation were detected.

2. On the possible cause of ESF semimonthly oscillation, the authors suggested that the planetary waves and/or lunar semidiurnal tide modulated the PRE which can play an important role on the ESF generation. This can be investigated further and demonstrated by using the simultaneous F layer height measurements from the SAO Luis digisonde. And also please explain in more detail on how lunar semidiurnal tides affect the PRE.

3. Regarding the identification of ESF start time, examples from both the airglow imager and VHF radar are suggested to be given. Further, as shown in Figure 1, the ESF structure is not obvious in the images taken at 23:48-00:21 UT. Please use arrows or other symbols to mark the ESF region.

4. "equatorial spread-F" and "equatorial plasma bubble" were used in the title/abstract and text respectively. For consistency please use "equatorial spread-F" or "equatorial plasma bubble".

5. How the oscillation amplitude was calculated, peak to trough? From Figure 3, the difference of ESF start time during the period is more than 2 hours, but the amplitude is ∼57 min.

There are some misprints in the manuscript. For example "did not allowed to", "it is well know that ". Please check the whole manuscript.
* * *

---

## Referee Comment (RC2) · Anonymous Referee #2 · 8 Jun 2019

This paper describes day-to-day changes in the onset time of the equatorial spread-F, as observed by an all sky imager and coherent back scatter radar in Brazil. It is found that the onset time occasionally shows a semimonthly (14.5d) variation. The authors present argument that the observed semimonthly variation could be due to the lunar semimonthly tide or 16d planetary wave.

I have two major concerns about this manuscript. Firstly, the introduction does not include sufficient information for the reader to understand which part of the results are new. It is stated in Page 6 Line 15, "Lunar semidiurnal tides have been pointed out as important factor to the appearance and the start time of EPBs" but there is no reference to it. If there are already such relevant studies, they must be properly cited, and more importantly, the authors should clarify what are the new results obtained in the present

study.

My second point is about the significance of the results. The authors fit a semimonthly (14.5d) curve to data segments that are sometimes shorter than one lunar cycle (Figure 2). I do not believe that it is appropriate to perform fitting to such sparse data unless the existence of the semimonthly variation is already known or highly expected. As the authors mentioned, the spread-F shows considerable day-to-day variability, and the authors' method could easily misinterpret random variability as a semimonthly oscillation. In my view there is no convincing evidence in this paper that supports the lunar semimonthly variation of the spread-F. The following are my comments on each event.

1. September 2003 (Figure 2a) There are only six data points. It is possible to fit "any" curve to such data. Thus the good fit does not necessarily suggest the semimonthly variation of the spread-F. The results actually seem to suggest that the start time of spread-F did not change much with time during this event.

2. October 2005 (Figure 2b) This is the most interesting event among those investigated in this paper. There is a shift in the start time of spread-F to later local times by almost two hours during 22-26 October and a shift to earlier local times during 29 October-3 November. Although it is not clear at this point whether these variations have anything to do with the lunar tide or 16d PW, this event deserves more detailed investigation. For instance, the authors could check whether the PRE plasma drift velocity shows consistent behavior. The authors should also examine whether source wave (lunar tide or 16d PW) existed in the middle atmosphere during this event.

3. November 2005 (Figure 2c) This has the same issue as the September 2003 event. The data are too few, so that the fitting is not reliable.

4. January 2008 (Figure 2d) The same as 1 and 3.

5. November 2005 (Figure 3) This is the same event as 3 (Figure 2c) but there is a
discrepancy in the phase of the semimonthly variation between Figure 2c and Figure 3. That is, the extension of the fitting curve in Figure 3 does not match the one in Figure 2c. This demonstrates the fitting technique used in this study is not reliable for extracting the semimonthly variation of spread-F.

6. November 2008 (Figure 4) The observed variation is very small (<30 min). The radar data is not able to resolve such a small variation as the authors mentioned.

As a summary, fitting a semidiurnal curve to small data segments is not a justifiable method to evaluate the influence of the lunar tide or 16d PW. This needs to be fixed before the paper is considered for publication.

What the authors could do instead is to take a statistical approach. Since the authors have long-term observations (September 2000 to December 2010), they could simply sort the data according to the lunar phase at the time of the observations, just like earlier researchers did to extract lunar tidal variations in other ionospheric parameters (e.g., Matsushita, 1967).

Matsushita, S., Lunar tides in the ionosphere, Handb. Phys., 49/2, 547, 1967

---

## Author Comment (AC1) · 2 Sep 2019

**Reviewer #1:**

REVIEWER:"**In this study, the authors investigated the start time of equatorial spread-F (ESF) by using the long term dataset (2000-2010) from an all-sky airglow imager and a coherent backscatter radar at the two stations Sao Joao do Cariri and Sao Luis, respectively. They reported that the semimonthly oscillations were seen in the start time of ESF dur- ing the periods September 2003, September-November 2005, January and November 2008. It was suggested that the ESF semimonthly oscillations could be associated with the 16-day planetary waves and/or lunar semidiurnal tides which affected the pre- reversal enhance-**

ment of eastward electric field (PRE) around sunset. The result of ESF semi-
monthly oscillation is interesting and deserves to be published."

AUTHORS: We are grateful for the dedicated time revising our paper. We have revised
the manuscript according to the reviewer's comments. We thank also for the language
revision.

"Although the authors stated that a long term dataset was used in this study, it
is not clear how often the semimonthly oscillation was observed. Whereas the
weather could cause lack of data from all-sky airglow imager, VHF radar should
not be affected. It would be better if one more figure can be included to provide
both the periods with data (all-sky airglow imager and VHF coherent, respec-
tively), and the periods when semimonthly oscillation were detected."

AUTHORS: The reviewer is right! As requested by the second reviewer as well. We
have included the statistical analysis for the all period of observations of the all sky
imager (Page 8 line 11 - page 9 line 11).

REVIEWER:"**On the possible cause of ESF semimonthly oscillation, the authors
suggested that the planetary waves and/or lunar semidiurnal tide modulated the
PRE which can play an important role on the ESF generation. This can be in-
vestigated further and demon- strated by using the simultaneous F layer height
measurements from the SAO Luis digisonde. And also please explain in more
detail on how lunar semidiurnal tides affect the PRE.**"

AUTHORS: We have requested those data to the INPE's colleagues. However, there
were coincident data only in three periods (Oct 2003, Oct 2005 and Nov 2005). Only in
early November there was observed a clear oscillation with such period in the time of
maximum vertical drifts of the F layer. We have included it in the manuscripts (Figure

3). Maybe the temporal resolution of the ionograms (10 min) is not enough to observed easily the oscillations aways. The lunar tide can change the start time of EPBs by modulating the wind that act driving the EXB vertical drift of the F region (Page 7 Lines 11-17).

REVIEWER:"**Regarding the identification of ESF start time, examples from both the airglow imager and VHF radar are suggested to be given. Further, as shown in Figure 1, the ESF structure is not obvious in the images taken at 23:48-00:21 UT. Please use arrows or other symbols to mark the ESF region.'**

AUTHORS: Thank you for the suggestion. We have tried to improve the visualization of Figure 1, but in the supplementary movie, the appearance fo the bubbles is clear.

REVIEWER:" **'equatorial spread-F' and 'equatorial plasma bubble' were used in the title/abstract and text respectively. For consistency please use 'equatorial spread-F' or 'equatorial plasma bubble'."**

AUTHORS: Thank you for the suggestion.

REVIEWER:"**How the oscillation amplitude was calculated, peak to trough? From Figure 3, the difference of ESF start time during the period is more than 2 hours, but the amplitude is ?57 min."**

AUTHORS: We have calculated it using the approach $start\ time\ =\ A\cos(\omega t\ +\ \phi)$, where $A$ is the amplitude, $\omega\ =\ 2\pi/14.5(days)$ and $\phi$ is the phase. The reviewer is correct, the difference between the minimum and maximum in Figure 3 is $\sim 2$ hours, which correspond to the amplitude of $\sim 1$ hour as shown in the chart.

[Figure]

REVIEWER:"**There are some misprints in the manuscript. For example ?did not allowed to?, ?it is well know that ?. Please check the whole manuscript.**"

AUTHORS: Thank you for the suggestion. We have revised the language as well.

―――――――――――――――――

---

## Author Comment (AC2) · 2 Sep 2019

REVIEWER:"**This paper describes day-to-day changes in the onset time of the equatorial spread-F, as observed by an all sky imager and coherent back scatter radar in Brazil. It is found that the onset time occasionally shows a semimonthly (14.5d) variation. The authors present argument that the observed semimonthly variation could be due to the lunar semimonthly tide or 16d planetary wave.**"

AUTHORS: We appreciate the revision and the contributions from the reviewer # 2. We have done our best to address all concerns pointed out by the reviewer.

REVIEWER:"**I have two major concerns about this manuscript.  Firstly, the in-**

[Figure]

**troduction does not include sufficient information for the reader to understand which part of the results are new. It is stated in Page 6 Line 15, "Lunar semidiurnal tides have been pointed out as important factor to the appearance and the start time of EPBs" but there is no reference to it. If there are already such relevant studies, they must be properly cited, and more importantly, the authors should clarify what are the new results obtained in the present study.**

AUTHORS: The reviewer is right. We have improved the introduction emphasising that this is the first time that kind of study was done using airglow images and corroborating with backscatter radar measurements. We have added the citation (Page 7, line 11) and some works have been cited though the manuscript.

REVIEWER:"**My second point is about the significance of the results. The authors fit a semimonthly (14.5d) curve to data segments that are sometimes shorter than one lunar cycle (Figure 2). I do not believe that it is appropriate to perform fitting to such sparse data unless the existence of the semimonthly variation is already known or highly expected. As the authors mentioned, the spread-F shows considerable day-to-day variability, and the authors' method could easily misinterpret random variability as a semimonthly oscillation. In my view there is no convincing evidence in this paper that supports the lunar semimonthly variation of the spread-F. **"

AUTHORS: According to the reviewer suggestion. We have included statistical analysis for the start time of EPB and the results showed that the influence of the semimonthly oscillation if frequent. It helps us to improve our discussion as well (Page 8 line 11 - page 9 line 11). We appreciate this comment from the reviewer.

REVIEWER:"**September 2003 (Figure 2a) There are only six data points. It is**

possible to fit "any" curve to such data. Thus the good fit does not necessarily suggest the semimonthly variation of the spread-F. The results actually seem to suggest that the start time of spread-F did not change much with time during this event.'

AUTHORS: In part, we agree with the reviewer. Any curve can be fitted to any data set. However, it fits very well and there is scientific reasons to set this kind of fit in the data. Please, see as following, another example, in which, the 14.5 days does not fit very well. Moreover, the statistical analysis can support those case studies showed in Figure 2 of the manuscript.

REVIEWER:"**October 2005 (Figure 2b) This is the most interesting event among those investi- gated in this paper. There is a shift in the start time of spread-F to later local times by almost two hours during 22-26 October and a shift to earlier local times during 29 October-3 November. Although it is not clear at this point whether these variations have anything to do with the lunar tide or 16d PW, this event deserves more detailed investigation. For instance, the authors could check whether the PRE plasma drift velocity shows consistent behavior. The authors should also examine whether source wave (lunar tide or 16d PW) existed in the middle atmosphere during this event.**"

AUTHORS: We thank the suggestions from the reviewer. We have checked the time of maximum vertical drift of the F layer from the digisonde data of São Luís. The results does not matched to the October, however in early November there was a clear oscillation. As explained to the Reviewer #1, the temporal resolution of the measurements could not be enough to observed this kind os variation.

REVIEWER:"**November 2005 (Figure 2c) This has the same issue as the Septem-**

[Figure]

**ber 2003 event. The data are too few, so that the fitting is not reliable.**"

AUTHORS: Same comment for Figure 2a.

REVIEWER: **"January 2008 (Figure 2d) The same as 1 and 3."**

AUTHORS: Same comment for Figure 2a.

REVIEWER:**"November 2005 (Figure 3) This is the same event as 3 (Figure 2c) but there is a discrepancy in the phase of the semimonthly variation between Figure 2c and Figure 3. That is, the extension of the fitting curve in Figure 3 does not match the one in Figure 2c. This demonstrates the fitting technique used in this study is not reliable for extracting the semimonthly variation of spread-F."** **"November 2008 (Figure 4) The observed variation is very small (<30 min). The radar data is not able to resolve such a small variation as the authors mentioned."**

AUTHORS: We agree with the reviewer that the technique used to analysed the data from the backscatter radar is not well resolved. We have decided to exclude this analysis from the paper and focus on the results from the airglow as suggested by the reviewer. We are working in the data of the radar to improve our analysis and results and we will present it in a future paper.

REVIEWER:**"As a summary, fitting a semidiurnal curve to small data segments is not a justifiable method to evaluate the influence of the lunar tide or 16d PW. This needs to be fixed before the paper is considered for publication."**

AUTHORS: We appreciate all the comments form the Reviewer # 2 and we are presenting a new version of the manuscript considered the main suggestions presented

here.

REVIEWER:"**What the authors could do instead is to take a statistical approach. Since the authors have long-term observations (September 2000 to December 2010), they could simply sort the data according to the lunar phase at the time of the observations, just like earlier researchers did to extract lunar tidal variations in other ionospheric parameters (e.g., Matsushita, 1967). Matsushita, S., Lunar tides in the ionosphere, Handb. Phys., 49/2, 547, 1967**"

AUTHORS: We have done it for the airglow data. Thank you for the suggestion.

**Fig. 1.** Same as Figure 3 of the manuscript, but for October 2005.

**Fig. 2.**

---

## Author Comment (AC3) · 2 Sep 2019

The comment was uploaded in the form of a supplement:
https://www.ann-geophys-discuss.net/angeo-2019-62/angeo-2019-62-AC3-supplement.pdf

---

## Author Comment (AC4) · 2 Sep 2019

The comment was uploaded in the form of a supplement:
https://www.ann-geophys-discuss.net/angeo-2019-62/angeo-2019-62-AC4-supplement.pdf

---

## Author Comment (AC5) · 2 Sep 2019

[revised manuscript text omitted]
 local lunar time for all period of observation of the all sky airglow camera. Note that only the start time of EPBs around of the sunset was considered, i.e., plasma bubbles that appeared in the airglow mages no later than 21:10 solar local time.

The lunar time was calculated as $\tau = t - \nu$, where, $t$ is the local solar time and $\nu$ is the age of the Moon, which depends on the phases of the Moon. Further details about the calculation of the lunar time can be found in Paulino et al. (2017) and references therein. In Figure 4, solid line represents the best fit for a 12 hours oscillation, which released an amplitude of $\sim 11 min$ and standard deviation of the fitted curve is shown by the dashed lines.

[Figure]

**Figure 4.** Start time of the EPBs (local time) as function of the lunar local time for whole period of observations.

[revised manuscript text omitted]

---

## Author Comment (AC6) · 2 Sep 2019

[revised manuscript text omitted]
 local lunar time for all period of observation of the all sky airglow camera. Note that only the start time of EPBs around of the sunset was considered, i.e., plasma bubbles that appeared in the airglow mages no later than 21:10 solar local time.

~~November 2008, one can observe two complete cycles of the start time of the BTSF fitting a semi-month oscillation with an amplitude of ~ 12 min.~~

The lunar time was calculated as $\tau = t - \nu$, where, $t$ is the local solar time and $\nu$ is the age of the Moon, which depends on the phases of the Moon. Further details about the calculation of the lunar time can be found in Paulino et al. (2017) and references therein. In Figure 4, solid line represents the best fit for a 12 hours oscillation, which released an amplitude of $\sim 11min$ and standard deviation of the fitted curve is shown by the dashed lines.

[Figure]

**Figure 4.** Start time of the EPBs (local time) as function of the lunar local time for whole period of observations.

[revised manuscript text omitted]

**List of changes**

---

## Author Response (AR2)

**Reply to the Editor and Reviewer**

**General Comments:**

Dear Dr. Fabiano Rodrigues.

First of all, thank you for consider the first round of revision of our paper. We also thank to the reviewers for dedication of time in reading and correcting our paper again. We have done our best to reply properly the concerns of the reviewer #2. Our point-by-point responses are listed as following and the necessary modifications were highlighted in the revised manuscript.

**Reviewer #2:**

REVIEWER:"**In the first round, I suggested that the authors should (1) improve the introduction to clarify which parts of the results are new, and (2) confirm the existence of the lunar tidal variation statistically, because fitting a semimonthly (14.5d) curve to small data segments does not prove the lunar tidal influence. I see that the authors have improved the manuscript. Concerning (1), I am satisfied with the introduction of the revised manuscript.**"

AUTHORS: We thank to the reviewer for considere our efforts in address the first concerning.

REVIEWER:"**Concerning (2), the authors responded by including a new figure that shows the lunar-time dependence of EPB based on all the available data (Figure 4). In Figure 4, the authors also show a fit of the semimonthly curve with one standard deviation and claim that the semimonthly variation in the EPB onset time is statistically significant. However, the statistical significance of the result is not clear from the figure or text. The amplitude of the semimonthly variation is 10.82 min, while the standard deviation is 20.12 min. This seems to suggest that there is actually no evidence for the semimonthly variation in the EPB onset time. I suggest that the authors should explain how the statistical significance was evaluated. Also, the authors could improve Figure 4 so that the reader can see the lunar tidal variation. One way to do that is to calculate the mean value at each hourly interval of lunar time and plot the mean values against lunar time, together with their corresponding standard error of the mean. See, for example, Figure 2 of Forbes et al. (2013). Forbes, J. M., Zhang, X., Bruinsma, S., and Oberheide, J. (2013). Lunar semidiurnal tide in the thermosphere under solar minimum conditions. Journal of Geophysical Research: Space Physics, 118(4), 1788-1801.**"

AUTHORS: The reviewer is right! As the text as Figure 4 were confuse. Again thank you for this important comment and contribution. We have replaced Figure 4 by a new one according to the suggestion of the reviewer and we have also included the suggested citation in the manuscript. We have also improved the text in order to explain the statistical significance of the results. So, The error bars represent the standard deviation of the mean and, of course, it can be higher than the mean values without loss of statistical significance. We have calculated the standard deviation to the amplitude and phase of the sinusoidal fit to the hourly binned points and found 4.8 min and 0.6 hours, respectively.

If we assume that it represents the measurements of uncertainties, 68% of the used hourly binned points must be between $\pm 1\sigma$ levels. Furthermore, the 12 hours fit is relatively significant because even we have 
[revised manuscript text omitted]

**List of changes**

---

## Author Response (AR3)

**Reply to the Editor**

**General Comment:**

Dear Dr. Fabiano Rodrigues.

We appreciate the efforts form you and Reviewer # 2 in order to help us improving this manuscript. We have incorporated all suggestions from the reviewer.

**Reviewer #2:**

REVIEWER:"**I1. Abstract - "the lunar tide, which has semimonthly variability, must be the main forcing" In my opinion, "must be" is too strong for the level of evidence presented in this study. "is likely" would be fine. 2. Abstract - I suggest that the authors remove "certainly" or replace it with "can". The importance of the lunar tidal contribution to the day-to-day variability of equatorial plasma bubbles depends on what aspect of equatorial plasma bubbles are considered. 3. Page 3 -The following sentence is not understandable: "Besides, the results shown here, in other period of observation was observed a tendency of the start time replaced to follow these periodicities." 4. Page 7 - Remove "which is, in fact, the maximum in the PRE" because this repeats the information. 5. Figure 4 - It should be mentioned somewhere in the text that Figure 4 is derived from all-sky imager data.**"

AUTHORS: Thank you again for the careful revision and suggestions. We have incorporated all of them in the text.

[revised manuscript text omitted]

**List of changes**